# The Role of the Microbiome on the Pathogenesis and Treatment of Colorectal Cancer

**DOI:** 10.3390/cancers14225685

**Published:** 2022-11-19

**Authors:** Irene Yu, Rongrong Wu, Yoshihisa Tokumaru, Krista P. Terracina, Kazuaki Takabe

**Affiliations:** 1Department of Surgical Oncology, Roswell Park Comprehensive Cancer Center, Buffalo, NY 14263, USA; 2Department of Surgery, Jacobs School of Medicine and Biomedical Sciences, State University of New York, Buffalo, NY 14203, USA; 3Department of Surgery, University of Florida, Gainesville, FL 32610, USA; 4Department of Gastroenterological Surgery, Yokohama City University Graduate School of Medicine, Yokohama 236-0004, Japan; 5Department of Surgery, Niigata University Graduate School of Medical and Dental Sciences, Niigata 951-8510, Japan; 6Department of Breast Surgery and Oncology, Tokyo Medical University, Tokyo 160-8402, Japan; 7Department of Breast Surgery, Fukushima Medical University, Fukushima 960-1295, Japan

**Keywords:** colorectal cancer, cancer, microbiome, tumor microenvironment, dysbiosis, microbes, immunotherapy, chemotherapy, carcinogenesis, prevention, diagnosis

## Abstract

**Simple Summary:**

The gut microbiome has recently become a hot topic for researchers looking to understand colorectal cancer carcinogenesis and progression. Understanding the role of the microbiome on colorectal cancer will allow us to better understand how to apply various preventative measures and therapies, including newer treatments such as immunotherapy. The aim of this manuscript is to provide a comprehensive review of the topic and explore the most recent updates on microbial mechanisms associated with the pathogenesis and progression of colorectal cancer, as well as their implications for prevention, diagnosis, treatment, and future research.

**Abstract:**

The gut microbiome has long been known to play a role in various aspects of health modulation, including the pathogenesis of colorectal cancer (CRC). With immunotherapy recently emerging as a successful treatment in microsatellite instability high (MSI-high) CRC, and with a newly demonstrated involvement of the gut microbiome in the modulation of therapeutic responses, there has been an explosion of research into the mechanisms of microbial effects on CRC. Harnessing and reprogramming the microbiome may allow for the expansion of these successes to broader categories of CRC, the prevention of CRC in high-risk patients, and the enhancement of standard treatments. In this review, we pull together both well-documented phenomena and recent discoveries that pertain to the microbiome and CRC. We explore the microbial mechanisms associated with CRC pathogenesis and progression, recent advancements in CRC systemic therapy, potential options for diagnosis and prevention, as well as directions for future research.

## 1. Introduction

In the United States, colorectal cancer (CRC) is the 4th most common cancer, falling behind only breast, prostate, and lung. It represents 7.9% of all new cancer cases. The vast majority of diagnoses are made between the ages of 65–74, although recently there has been an increasing number of diagnoses in younger patients. Due to improvements in screening methods, the majority of patients have only local and/or regional disease at the time of diagnosis, but those that do have metastatic disease still make up over 20% of cases [1].

Loco-regional disease is often treated with upfront surgery, aiming to cure. Patients with a resectable primary who also have resectable liver or lung metastases should undergo surgical resection of both sites with curative intent. For patients with advanced or widely metastatic disease, the standard of care is systemic chemotherapy with or without radiation therapy. A subset of patients qualify for neoadjuvant therapy to improve disease-free survival and possibly downstage the disease to make an unresectable tumor resectable [2]. There has been a growing role for immunotherapy in CRC, given the recent successes in patients with melanoma, lung cancer, and renal cancer. Many studies have attempted to investigate possible avenues by which we can improve the efficacy of immunotherapy, as well as the chemotherapy and radiation regimens we already have so as to better treat patients with advanced presentations. 

A growing body of literature examines how the various microbiomes of the human body, especially the gut microbiome, play a role in the process of carcinogenesis and the metastatic spread of cancer, as well as their impacts on the efficacy of CRC treatment. Recent breakthroughs in genomic sequencing technology now allow the identification of microorganisms that were previously impossible to detect with bacterial culture. With the elucidation of these processes, novel concepts arising from these previously obscured interactions can now be applied to improve the prevention, screening, and diagnostic markers of CRC. In this review, we examine both what is known and under investigation regarding the role of the microbiome on CRC pathogenesis and progression (Figure 1). We explore the clinical applications of this knowledge, including recent advancements in CRC systemic therapy, options for diagnosis and prevention, as well as directions for future research.

## 2. Microbiome and CRC Pathogenesis

Knowledge of the microbiome-mediated mechanisms of CRC pathogenesis is continuing to evolve, with ongoing research examining the complex interaction between microbes and host cells that contributes to tumor formation and progression [3]. Emerging CRC therapies target these mechanisms, but a more comprehensive understanding of this aspect of carcinogenesis is necessary to transform the prevention and management of CRC.

### 2.1. Dysbiosis and Colorectal Cancer

The human gastrointestinal microbiome contains greater than 1000 species of bacteria, mostly of Bacteroidetes and Firmicutes phyla in healthy individuals [4,5], with *Ruminococcus*, *Bacteroides*, and *Prevotella* dominant at the genus level [6]. It has been hypothesized that one of the mechanisms through which some of these bacterial genera protect against CRC is through the prevention of pathogenic bacteria and fungi overgrowth [7]. The balance of microbes can undergo significant alterations resulting in major changes to the overall composition of the microbiome. This unfavorably altered composition is coined “dysbiosis” [8,9].

The microbiomes of patients with CRC have been associated with significant dysbiosis when compared to the microbiomes of healthy patients [5]. A non-exhaustive list of bacteria, thought to contribute to CRC pathogenesis, includes *Fusobacterium nucleatum* [10,11], *Escherichia coli* [9,12], and enterotoxic *Bacteroides fragilis* [13], with the concurrent loss of certain gut commensal bacteria such as *Bifidobactium animalis* and *Streptococcus thermophiles* [9]. Given these findings, dysbiosis has been hypothesized to play a key role, and possibly a causative one, in CRC pathogenesis, especially considering that many of the well-established risk factors for CRC, such as obesity, inflammatory bowel disease, and high fat and protein diets, are also linked to intestinal dysbiosis [14]. It is important to note, however, that although there have been many studies identifying specific bacterial species both overrepresented and underrepresented in intestinal dysbiosis, it is still unclear whether these changes are truly causative of disease processes. Some have suggested that factors often associated with dysbiosis, such as chronic inflammation, are not necessarily a result of the dysbiosis, but may, in fact, be causing it [15]. Because many patients also have multiple comorbidities that have been associated with dysbiosis, one must consider the dysbiosis in relation to the host’s overall state of health and assume that alterations to the microbiome are reflective of all these comorbidities. Until we have a better understanding of these complex interplays, caution must be taken in attributing the presence or absence of any one specific bacterial species to a specific disease.

Still, it has been shown that several factors can affect the composition of colonic microbiota, including lifestyle, diet, acquisition of new commensals, and drug uptake, thus disrupting the host-microbiota homeostasis [11]. Alcohol consumption, for example, has been shown to deplete bacteria that exert anti-inflammatory activity, resulting in damage to the colonic wall [16]. Oral bacteria, which come from a distinct microbiome, have also been shown to disseminate into the colon, altering the colonic microbe composition [11]. Under the influence of factors such as these, dysbiosis occurs via a combination of the following changes: (1) a reduction in commensal bacteria, (2) the overgrowth of opportunistic pathogens, and (3) a decrease in the overall diversity of the microbiome [14].

Though investigations on dysbiosis in the setting of CRC have mostly focused on the bacterial composition of the microbiome, a few studies have looked at the contribution of alterations to the fecal virome [17,18] and mycobiome [19,20]. Human papillomaviruses (HPV) are known to play a role in the formation of head and neck, cervical, genital, and anal cancers. There is emerging evidence for a role in CRC, especially those of the rectum, an area of concern due to the increasing incidence of rectal cancers in younger cohorts of patients [21], without obvious genomic alterations from average-onset CRC [22]. For example, a 2010 epidemiological study from Taiwan that utilized a cohort of 10,612 patients tracked through a cancer screening program demonstrated a significant increase in CRC incidence in females who were found to be HPV-positive with strains other than the low-risk HPV 6 and 11 strains [23]. Interestingly, despite the p53 mutation that occurs along the standard CRC mutation pathway, HPV-infected CRC cells were found to have an intact p53 gene, suggesting that HPV could induce CRC carcinogenesis by disabling the functionality of this protein [24]. 

There have also been numerous studies investigating the role of the Epstein–Barr virus (EBV) in CRC carcinogenesis because of its presence in tissues affected by diseases such as ulcerative colitis [25], although to date no studies have conclusively supported this link. Others have studied the role of human papillomavirus (HPV) in congruence with EBV-mediated CRC pathogenesis, again without conclusive results [26]. Overall, this limited data suggests that dysbiosis of the gut virome may very well be associated with the development of CRC [18], although causality has yet to be proven since investigations into dysbiosis of the gut virome and mycobiome are still in their infancy.

### 2.2. Inflammation and Toxins from Dysbiosis

Once dysbiosis occurs, the colon is subject to chronic inflammation, induced through a variety of mechanisms including the depletion of anti-inflammatory bacteria, secretion of toxins, and synthesis of metabolic byproducts, such as acetaldehyde and N-nitroso compounds [7]. 

Butyrate, for example, is a short-chain fatty acid (SCFA) that is also created as a byproduct by many colonic bacteria. It serves as a source of energy for colonic enterocytes, but has also been shown to induce dysfunction in the epithelial barrier, in turn causing inflammation by activating interleukin-6 (IL-6), cytokines, and tumor necrosis factor-α (TNF-α) [7], particularly when present in high concentrations [27]. In fact, many groups, including our own, have shown that IL-6 and TNF-α cause an amplification loop of chronic inflammation in colon carcinogenesis [28]. Interestingly, there are also many studies that advocate for butyrate, along with other SCFAs, as mediators against inflammation by downregulation of inflammatory cytokines and upregulation of colonic regulatory T cells (Treg), which play a key role in the suppression of inflammation [29,30,31,32]. This suggests that the effects of butyrate are concentration-dependent, and that regulation to within a certain concentration may be necessary to maintain an anti-tumorigenic and anti-inflammatory effect.

Secondary bile acids, which result from bacterial fermentation of the primary bile acids secreted by the liver and gallbladder during fat digestion, have been associated with tumor formation and are thought to be carcinogenic as well [33,34,35]. Indeed, we have reported that bile acids function as signaling molecules in multiple settings [36]. Proposed mechanisms include the triggering of oxidative damage and mitochondrial dysfunction, which leads to the growth of the tumor [37]. 

Additionally, certain bacteria produce toxins that directly damage epithelial cells. For example, enterotoxic *Bacteroides fragilis* produces the enterotoxin fragylisin, which enhances cytokine secretion [38] and cleaves E-cadherin on colonocytes, leading to increased mucosal permeability [39]. *Fusobacterium nucleatum*, while not known to produce enterotoxins, similarly can induce inflammatory changes by directly invading epithelial cells via FadA surface protein interaction with E-cadherin, resulting in accelerated cancer cell proliferation [40]. *Bacteroides fragilis* and *Enterococcus faecalis* both produce reactive oxygen species (ROS) causing oxidative damage [7,41]. *Escherichia coli* and *Klebsiella pneumonia* have been shown to produce the genotoxin colibactin, which some have suggested could serve as a biomarker of CRC [42,43].

The cumulative inflammation that results from these mechanisms then creates an environment in which otherwise non-pathogenic bacteria with genotoxic potential are able to adhere to the intestinal wall and induce tumorigenesis [44].

### 2.3. Evidence from Biofilm

Biofilm is a term used to describe the organizational and structural components of a microbiome. Normal colonic epithelium in healthy individuals is typically coated in a layer of sterile mucus with effective compartmentalization of bacteria to the lumen [7]. Biofilm formation represents a change in the epithelium-microbiome interaction. This change provides a resilient structure that decreases the penetration of antibiotics, increases epithelial permeability via loss of E-cadherin, and allows continued interaction between tumor-promoting bacteria and host cells [8,20,45]. This continued interaction at the level of unprotected epithelial cells promotes pro-carcinogenic tissue inflammation, inducing mutations within the epithelial cells, resulting in the increased proliferation seen with the growth of tumors [45]. These findings are present both in cases of sporadic CRC, as well as in certain genetic syndromes, such as familial adenomatous polyposis (FAP) [46].

Patients with CRC can be broadly divided into two different groups: biofilm-positive and biofilm-negative. A study by Dejea et al. demonstrated that among the subset of biofilm-positive patients, polymicrobial biofilms are present nearly universally on tumors of the proximal colon, but less so on distal tumors [45]. Regardless of tumor location, however, patients who demonstrated biofilm-positive tumors also had biofilms on tumor-free mucosa, at locations separate and distant from their tumors [45]. They concluded that the risk of developing CRC is a striking five times higher in biofilm-positive patients compared to those who were biofilm-negative [45].

Animal studies have shown that homogenates of biofilm-positive colon mucosa taken from human patients with colorectal tumors induced colon tumor formation in mice, whereas homogenates from biofilm-negative patients did not. Additionally, biofilm-positive homogenates from patients without colon tumors also induced colon inflammation and tumors in mice, suggesting that biofilms, both from hosts with and without colon tumors, are carcinogenic, at least in murine models of CRC [47].

### 2.4. Dysbiosis and Tumor Microenvironment

The tumor microenvironment (TME) is defined as the environment surrounding the cancer cells within the bulk tumor [48]. This includes infiltrating immune and inflammatory cells, blood and lymphatic vessels, the extracellular matrix, stromal cells such as fibroblasts and adipocytes, and secreted proteins [49]. There is a large body of growing research that aims to further our understanding of the mechanisms within the TME that promote tumor growth, invasion, and metastasis, because these factors heavily impact both prognosis and the effectiveness of anti-cancer therapies [50,51,52]. Although these mechanisms are not well-understood, there are studies supporting the concept that intestinal bacteria exert influence on the TME, for example, through the regulation of the inflammatory response [53].

Tumor-associated macrophages (TAMs) are a type of tumor-infiltrating immune cell [54,55]. TAMs exhibit two relevant phenotypes that were first coined by Mills et al. in 2000 as M1 and M2 [56]. Since then, these descriptors have been used in various ways, and the nomenclature remains vague [57]. For our purposes, we present what appears to be the most common use of this nomenclature in CRC: M1 macrophages are pro-inflammatory but tumor-preventing [58,59,60,61] and M2 macrophages are anti-inflammatory yet tumor-promoting [58,62,63]. As such, decreases in M1 macrophages, which can be caused by variations in intestinal microbiota, are associated with a reduced immune response to tumors [64]. Conversely, increased M1 macrophage infiltration has been associated with an improved prognosis in CRC patients, despite a surprising concomitant increase in M2 macrophages as well [62]. The role of TAMs appears context dependent, which may be due to variations in detection methods [55].

Tumor-associated neutrophils (TANs) [65] have similarly been suggested to have two phenotypes: N1 neutrophils that are tumor-preventing, and N2 neutrophils that are tumor-promoting. TANs exhibit plasticity between these two phenotypes depending on signals from the TME. Interferon (IFN)-β and transforming growth factor (TGF)-β signaling may play a role in regulating this plasticity, but the exact surface markers and transcriptional regulators that are associated with these two phenotypes are still under investigation [66]. N2 TANs have been shown to promote angiogenesis and tumor invasion, and are also involved in metastasis by means of neutrophil extracellular traps (NETs) [66]. NETs are created when neutrophils expel their intracellular contents to create scaffolds of DNA and chromatin, meant to capture and kill pathogens [67,68]. NETs also capture circulating cancer cells, awakening those that are dormant [69]; it is in this way that N2 TANs promote tumor invasion and metastasis [67,68,70]. There are many stimuli that induce NET formation, including the presence of inflammatory cytokines, certain metabolites, bacterial toxins, and ROS [67], all of which, as discussed above, are influenced by changes in the microbiome, although a link between dysbiosis and NET formation in cancer has yet to be directly investigated. Interestingly, studies have shown that although high levels of intratumoral TANs are associated with shorter disease-free survival (DFS) and overall survival (OS), certain TANs, specifically CD66b+ cells, may be associated with a better prognosis in CRC [71]. Peritumoral and stromal TANs were not associated with any significantly shorter DFS or OS [72], but studies have shown that stromal NET density can be associated with worse recurrence in certain types of cancer [73]. 

Tregs, as mentioned previously, are a subset of T-cells with the intracellular marker FoxP3 that have been shown to suppress inflammation and the immune response, thereby maintaining self-tolerance [74]. In the intestine, Tregs regulate homeostasis by limiting the number of inflammation-inducing CD4+ T cells, and this regulation appears to be improved in the presence of SCFAs [75]. Although one might expect that maintenance of homeostasis and suppression of inflammation may be protective against CRC, studies have shown that Treg upregulation in tumors predicts recurrence, whereas downregulation of Tregs in tumors is associated with longer periods of DFS and OS [64,76,77]. This may be due to the fact that increased self-tolerance within the tumor improves the escape from immune recognition, allowing the tumor to continue growing. On the other hand, there are certain subsets of Tregs that are thought to be protective, such as CD4CD8αα Tregs, which are found in abundance in healthy colonic mucosa [78]. CD4CD8αα Tregs demonstrate decreased activity when *Faecalibacterium prausnitzii*, a key butyrate-producing commensal in healthy adults [79] and member of the Firmicutes phylum, is reduced, suggesting that the decreased activity of CD4CD8αα Tregs may be involved in colonic carcinogenesis [80]. When our group quantified Tregs using the transcriptomic algorithm xCell, we discovered that an abundance of Tregs was associated with lower amounts of *Lachnoclostridium*, *Ornithobacterium,* and *Flavivirus* in the TME as well as with an improved response to the chemotherapy regimens of mFOLFOX6 and FOLFOX/FOLFIRI without bevacizumab [51]. Further studies are needed to clarify how the interplay between microbes and specific subsets of Tregs affect CRC carcinogenesis, as well as how the location of Tregs within the TME, whether they are intra-tumor, peri-tumor, or within the stroma, changes their effect.

TME stromal cells have also been shown to play a role in carcinogenesis. Cancer-associated fibroblasts (CAFs) [81], which are the most abundant non-cancer cell in a tumor, have been shown to enhance tumor growth and metastasis after attaining a ROS-induced catabolic phenotype. They do so by secreting growth factors and inflammatory ligands, as well as by creating a microenvironment rich in nutrients and mitochondrial fuels [82]. Melanoma studies have shown that CAFs may also interfere with the proper cytotoxic T lymphocyte elimination of cancer cells within the tumor [83], thus allowing further growth. On the other hand, CAFs were shown to be associated with better survival in pancreatic cancer [81].

Cancer-associated adipocytes (CATs) [84], another type of stromal cell in the TME, have been shown to promote carcinogenesis through obesity-associated hormones (e.g., leptin), vascular endothelial growth factor (VEGF), and cytokine effects on cancer-prone host cells [85,86]. Conversely, an abundance of CATs in the TME was associated with a less proliferative tumor type in hepatocellular carcinoma and breast cancer [87]. Given the role of cytokines and ROS in stromal cell effects and their regulation by intestinal dysbiosis, one could envision a link between gut microbiota and TME stromal cells, which would be worth investigating.

### 2.5. Mitochondria

Mitochondria are intracellular organelles that serve as energy production facilities for the cell. They can also induce apoptosis during times of cellular stress. Recently, they have been implicated in carcinogenesis by three mechanisms: (1) the generation of reactive oxygen species (ROS), (2) the accumulation of mitochondrial onco-metabolites resulting from genetic mutations, and (3) the resistance to mitochondrial-driven apoptosis [88,89,90,91]. It has been proposed that intestinal microbiota signal to gut epithelial cell mitochondria, inducing mutations in mitochondrial DNA that cause decreased oxidative phosphorylation capacity. While not necessary for cancer formation, one characteristic of cancer cells is increased rates of glycolysis at the expense of oxidative phosphorylation, also known as the Warburg effect. This shift towards glycolysis likely occurs due to the inability of disordered vasculature to provide enough oxygen to the rapidly growing tumor [92]. In cancer cells, however, glycolysis may also occur in the presence of oxygen due to altered cancer metabolism [93]. Hence these mitochondrial DNA mutations are thought to contribute to carcinogenesis [93,94]. Microbe-mitochondria signaling has also been shown to affect mitochondrial-induced apoptosis; bacteria may decrease the apoptosis of cancer cells, and viruses that inhibit apoptosis may directly contribute to the carcinogenesis of the cells they are infecting [94]. While microbe-mitochondrial crosstalk may drive some aspects of CRC carcinogenesis, these mechanisms are still not fully understood, and represent an area that should be further investigated.

## 3. Diagnosis and Treatment Implications

The effects of the microbiome and our interventions to alter it hold many implications for our management of CRC (Figure 2) in the realms of prevention, diagnosis, and treatment. The remainder of this paper will focus on these implications, starting with microbial effects on the diagnosis and treatment of CRC.

### 3.1. Prediction of Cancer Development 

Because bacteria have been shown to directly affect carcinogenesis, there has been a growing body of work looking into features of the microbiome along the adenoma-carcinoma sequence [95] and whether the detection of these features could be employed in screening and detection. It has been suggested, for example, that the detection of colonic mucosal biofilm formation may be helpful in predicting the development of sporadic CRC [45].

Studies have shown that Proteobacteria become progressively more prominent in patients with adenomas and even more so in those with carcinoma [96]. Multiple studies have shown that Fusobacteria is also prevalent in premalignant lesions of the colon, mostly adenomatous polyps [97,98,99,100]. A remarkable metagenomic study by Yachida et al. evaluated the microbiome composition at various stages of CRC development and found that there were significant shifts at particular stages [101]. Specifically, *Fusobacterium nucleatum* was found to be abundant in intramucosal carcinoma and across later stages of cancer development. On the other hand, the *Atopobium parvulum* and *Actinomyces odontolyticus* species appear specific to the presence of multiple polypoid adenomas and intramucosal carcinoma only [101]. Other fecal sequencing studies have demonstrated that *Bacteroides* appear enriched in patients who developed adenomas and could possibly play a role in adenoma formation [9]. Wirbel et al. performed a meta-analysis study of fecal metagenomes and identified a core set of 29 species associated with CRC, including the well-documented *Fusobacterium,* along with *Parvimonas, Gemella, Prevotella, Porphyromonas, Peptostreptococcus, Solobacterium,* and bacteria of the Clostridiales order [102]. Altogether, these studies indicate that the gut microbiome deteriorates along the adenoma-carcinoma sequence and suggest that certain characteristic metagenomic linkage groups could eventually serve as an alternative to colonoscopy for the diagnosis of these tumors. Unfortunately, however, no reliable predictive model that can differentiate healthy patients from those with adenomas has been developed and verified. More research in this regard is required [96]. 

### 3.2. Germline Mutations and Early-Onset CRC

As with many other hereditary cancers, germline mutations for CRC greatly increase an individual’s risk of developing cancer due to the two-hit theory: both copies of a tumor-suppressor gene need to be disabled for a cancer cell to develop. When one copy of the tumor-suppressor gene is disabled at baseline by the germline mutation, the patient only requires one sporadic mutation instead of two. The two most common hereditary CRC syndromes are Lynch Syndrome, in which there is a mutation in a mismatch repair gene, and Familial Adenomatous Polyposis (FAP), in which there is a mutation in the tumor-suppressor gene adenomatous polyposis coli (APC). 

There are only a few studies in the literature that attempted to identify microbiome changes associated with hereditary syndromes. Some bacterial species found to be associated with Lynch syndrome adenomas and malignant tumors include *Bacteroides* species, *Parabacteroides distasonis*, *Faecalibacterium prausnitzii*, *Ruminococcus bromii*, *Pseudomonadaceae* family, *Escherichia coli*, *Klebsiella* species, *Porphyromonas*, and *Methanobrevibacter* [103,104,105]. Interestingly, there did not appear to be any association with *Fusobacterium nucleatum* [103,104]. In FAP, there are associations with *Bacteroides fragilis* and *Escherichia coli* [46]. The role these microbes play in hereditary syndromes is unclear, and the lack of studies looking to understand this represents a massive gap in research and an opportunity to improve screening and prevention in these high-risk patients [106].

Although Lynch Syndrome and FAP predispose patients to CRC at a younger age, the increasing incidence of early-onset CRC cannot be explained by hereditary syndromes. This increase in incidence is likely multifactorial, and some suggest that the microbiome may be the key mediator [107]. There are some studies that seek to evaluate this theory, but the relationship between dysbiosis and early-onset cancer is underexplored [108]. Yang et al. identified *Flavonifractor plautii* as an important species in early-onset CRC [109], and Akimoto et al. suggest that *Bifidobacterium*, which is associated with signet ring cell formation often seen in early-onset CRC, may also play a role [110]. There are some studies suggesting that *Fusobacterium nucleatum* plays a role in the development of non-hereditary early-onset CRC, although unlike in late-onset CRC, this relationship is understudied [110], and the literature is inconsistent, with some studies also suggesting that there is no correlation [111,112]. Overall, exploring the interplay between the multiple environmental, microbial, and genetic risk factors that contribute to early-onset CRC and creating a reliable and well-tested risk score based on these factors could help identify at-risk individuals and allow them to start preventative measures or treatment on an earlier timeline [113].

### 3.3. Microbiome Effects on Chemotherapy

Chemotherapy is the backbone of systemic treatment for CRC. Multiple lines of evidence suggest that the intestinal microbiome can influence the efficacy of chemotherapy as well, through mechanisms such as modulating host immune response, affecting drug metabolism, mediating toxicity, and activating inflammatory pathways [114,115].

For example, oxaliplatin efficacy is mediated by intestinal microbiota affecting the recruitment of immune cells and intratumoral production of ROS [53]. On the other hand, *Fusobacterium nucleatum* has been shown to induce resistance to oxaliplatin and 5-fluorouracil (5-FU) via the promotion of autophagy in CRC cells [116]. Autophagy has been shown to be protective for tumor cells under hypoxic conditions [117], which as previously discussed, is common in tumors. Studies in germ-free and antibiotic-treated mice demonstrate lower immune cell response to therapy, again emphasizing the importance of microbiota and TME on the efficacy of chemotherapy [53].

Interestingly, certain ileal bacteria have been shown to improve the efficacy of oxaliplatin alone and in conjunction with immune checkpoint inhibitors, a type of immunotherapy. All gut bacteria have varying levels of immunogenicity, which is the ability to trigger a host immune response. Studies in mice have revealed that some bacteria are overrepresented in responders to chemotherapy and immunotherapy, suggesting they are highly immunogenic. These bacteria include *Propionibacterium acnes*, *Alistipes onderdonkii*, *Erysipeloclostridium ramosum*, *Eggerthella lenta*, and *Streptococcus anginosus* [118], which all appear to enhance oxaliplatin-induced ileal crypt epithelial cell apoptosis. This apoptosis in turn primes and activates helper T-cells, thus modulating the host immune response against cancer cells [118]. Similarly, the effects of cyclophosphamide have been found to rely on small intestine bacterial translocation into secondary lymphoid organs, thus stimulating helper T-cells [119]. As such, it has been suggested that within the small intestine, a proper balance of immunogenic and tolerogenic commensal bacteria and proper ileal cell apoptosis are two conditions that should be met in order to facilitate a good response to chemotherapy [6].

Microbes have also been found to mediate toxicity to chemotherapeutic agents, which if severe, can limit the patient from receiving therapeutic dosing. For example, deactivated Irinotecan is eliminated through the gastrointestinal tract, and severe treatment-limiting diarrhea occurs when the gut *Clostridium* species glucuronidate this deactivated Irinotecan into its active form [115,120].

Given the multitude of microbiome-induced effects on the therapeutic use and toxicities of chemotherapy, there is a high potential for modulation of the microbiome to improve both the efficacy and patient tolerance of chemotherapy.

### 3.4. Potentiation of Radiation Therapy

The microbiome and its relationship with the efficacy of radiation therapy have been less extensively examined than for chemotherapy, but it has been shown to mediate radiation-induced toxicities. Radiation enteritis is an example of toxicity from radiation that can significantly alter a patient’s quality of life due to long-term sequelae including stricture, fistula, perforation, bleeding, and abscess [121]. Studies in germ-free mice have shown that intestinal bacteria are essential for the development of radiation enteritis [122]. Symptoms such as post-radiation fatigue [123] and diarrhea [124] have also been linked to distinct microbial profiles when compared to the profiles of patients who did not experience these adverse effects. Studies from the small bowel have shown that microbiota may be protective against radiation-induced epithelial cell death by binding Toll-like receptors (TLRs) and activating the nuclear factor-kappa B (NF-κB) pathway, which was associated with elevated expression and activation of the P53 tumor suppressor [125]. Some have suggested that the use of strain-specific probiotics or antibiotics to modulate the microbiome may improve tolerance to radiation therapy by reducing adverse effects [126,127], but this is an area that still requires additional research.

### 3.5. Immunotherapy

Immune checkpoint inhibitors (ICIs) are a form of immunotherapy that blocks the tumor cells from downregulating the immune response against them [128]. The most prominent ICIs in use today are anti-PD1 and anti-CTLA4 drugs. In CRC, these medications are currently only approved for use in patients with documented deficiencies in mismatch repair (dMMR) and microsatellite instability (MSI) [129]. While some patients have been shown to achieve a complete response, such as the one CRC patient with dMMR/MSI in the landmark anti-PD1 pilot study [80], as well as all 12 advanced rectal cancer patients enrolled in a recent trial (NCT04165772) [130], these patients only comprise approximately 5–15% of patients with CRC, so this approach cannot be generalized to all the CRC patients. To this end, there has been a surge of interest regarding the role of the gut microbiome in improving ICI efficacy, with most studies focusing on melanoma, which has seen the most clinical use of ICIs. Nonetheless, these studies have demonstrated that the gut microbiome can alter the response to ICIs, with *Bifidobacterium* and *Faecalibacterium* facilitating anti-PD1 activity [131,132] and certain *Bacteroides* facilitating anti-CTLA4 activity [133]. Given these findings, some have proposed that developing a multiparameter model integrating microbiome composition, TME, and germline genetics may help clinicians predict who might best benefit from ICIs [134]. With continued advancements in research, the hope is that these theories could be applied to CRC as well.

The microbiome in the ileum has also come under investigation as a player in CRC response to ICIs; in fact, some argue that it may be the ileal microbiome instead of the colonic one that serves a central role in upregulating our immune response against cancer cells [6]. Roberti et al. found that there were synergistic effects of ICIs with the immunogenic chemotherapy agent oxaliplatin, and that immunogenic bacteria from ileal samples such as *Bacteroides fragilis* or *Erysipeloclostridium ramosum* improved responses [118]. On the other hand, tolerogenic bacteria such as *Paraprevotella clara* or *Fusobacterium nucleatum* actually blunted the effects of therapy [118].

Another entity that is currently being investigated as a target for immunotherapy is CD47, the presence of which on tumor cells allows them to escape immune clearance by signaling to macrophages that they are not targets for phagocytosis [20,135]. Animal studies utilizing anti-CD47 immunotherapy have shown that non-responder mice with colon tumors convert to responders when fed live *Bifidobacterium*, due to microbe localization to the TME, facilitating the response via STING signaling [136], which is quite promising. 

Tumor vaccines are a type of immunotherapy that aim to enhance the memory of the immune system in order to prevent recurrence. As previously discussed, via crosstalk between tumor cells, immune cells, and microbes within the TME, tumors can facilitate their own growth and escape immune recognition. There are many types of vaccines currently under investigation, including molecular-based, cell-based, and vector-based vaccines, which all target those specific interactions between elements of the TME [137]. The vaccines activate immune cells, such as TAMs, after they are exposed to specific tumor-associated antigens (TAAs) or tumor-specific antigens (TSAs), causing them to attack malignant cells [129]. Identifying the antigens that are specific to a patient’s tumor is both time and resource intensive [129], but once identified, they can be used to tailor cancer treatment to the individual patient. There are over 1000 clinical trials currently investigating the potential clinical use of TAAs and TSAs [138].

Interestingly, there has also been a growing area of research regarding blood, which typically is considered sterile, and the presence of a healthy human blood microbiome [139]. A recent study by Yang et al. demonstrated that patients with CRC who were identified as responders to combined adoptive T-cell immunotherapy and chemotherapy had a more diverse blood microbiome, including *Bifidobacterium*, *Lactobacillus*, and *Enterococcus* compared to non-responders [140]. This suggests that the composition of the blood microbiome has the potential to serve as a predictive marker for clinical responses to systemic treatment for CRC. Yet in response to the studies examining the possibility of a blood microbiome, which were mostly either small-scale or without the ability to truly distinguish between contaminants and the presence of true bacterial commensals, there have also been some studies suggesting the opposite. Tan et al. performed an analysis on blood samples of 9770 humans and concluded that there is no consistent endogenous human blood microbiome [141]. Instead, they suggest that the presence of asymptomatic blood bacteria is the result of the transient translocation of commensal bacteria from other sites with endogenous microbiomes, such as the gut [141]. Because responders to immunotherapy and chemotherapy have been associated with certain commensal bacteria, such as *Bifidobacterium*, this transient translocation could also explain the presence of such commensal bacteria in the blood of the responders reported in the study by Yang et al. [140]. It is clear that further research is needed in this field.

### 3.6. Neoadjuvant Immunotherapy and Surgery

With all the progress being made in immunotherapy, there has also been a growing body of research examining the benefits of using it as a neoadjuvant treatment. It has been suggested that neoadjuvant immunotherapy can generate protective immunity, allowing the body to identify and remove micrometastases, a common cause of recurrence after surgical resection [142]. Common adverse effects of ICIs, such as thyroid dysfunction, fatigue, diarrhea, and hepatitis, could also be minimized by the use of a comparatively short and directed neoadjuvant course [129]. If neoadjuvant therapy is successful, certain patients with locally advanced forms of CRC could be considered for curative surgery, whereas surgery otherwise may have been precluded, for reasons such as severe morbidity associated with an en bloc resection or inability to cure. 

The NICHE trial demonstrated that in a cohort of 40 patients, half of whom had dMMR CRC and half of whom had mismatch repair proficient (pMMR) CRC, the combination of ipilimumab and nivolumab resulted in an impressive 100% pathologic response in dMMR CRC. Of these, 95% demonstrated a major pathologic response, defined as less than or equal to 10% residual tumor, and 60% had a complete pathologic response. Interestingly, in patients with pMMR, for whom ICIs have been less efficacious, 27% still demonstrated a pathological response [143]. The subsequent NICHE-2 trial was performed in a larger cohort of 112 dMMR CRC patients with similar results: 95% of patients had a major pathologic response, and 67% of all patients had a complete pathologic response [144]. The median time to surgery after the first dose of neoadjuvant immunotherapy was approximately 5 weeks. 

Similarly, a recent study by Zhang et al. examined the use of neoadjuvant immunotherapy in six patients with dMMR/MSI gastrointestinal tumors, including two with CRC, who were at high risk of recurrence were they to undergo only surgery [145]. All six patients demonstrated pathologic responses to the neoadjuvant treatment, with five out of the six patients demonstrating complete response. All operations following neoadjuvant immunotherapy were performed safely [145]. Furthermore, the NCT04165772 trial demonstrated that 100% of the advanced dMMR/MSI rectal cancer patients achieved a pathological complete response after neoadjuvant ICI [130], which will change the management of these patients. These results suggest that we may be entering an era where pinpointing a specific group of patients that respond completely to a targeted treatment will be the standard, instead of looking for a silver bullet that will treat the majority of patients. By investigating more strategies to improve immunotherapy efficacy, including the alteration of the microbiome, perhaps additional therapies that will achieve a complete or major pathologic response can be identified, allowing more patients to receive the benefits. 

### 3.7. Precision and Personalized Medicine

A common theme that has come up in recent literature and across this review is the idea of precision medicine, also known as personalized medicine, in CRC. This stems from the understanding that there are many individual factors contributing to the pathogenesis and development of CRC that are unique to the individual. As discussed above, these factors include patient genetics, the individual’s healthy microbiome, dysbiosis as a departure from that patient’s “normal” and healthy microbiome, and environmental influences such as diet, among others. The need for this approach is further emphasized by the recent advances in immunotherapy that appear most efficacious in the select group of dMMR CRC. Being able to tailor treatment plans to the patient’s individual circumstance and tumor may in fact be the best way to optimize a patient’s treatment for efficacy and tolerance.

Molecular pathological epidemiology (MPE) is a field of research that, as the name suggests, integrates pathology and epidemiology in the form of data science to further our understanding of pathology at the molecular level and how this translates to clinical outcomes [146]. We are slowly starting to see an increase in the use of MPE as a means to understand the complex interplay between the multiple internal and external factors that contribute to CRC pathogenesis and development. For example, Chen et al. performed a very recent analysis of the fecal metagenome and metabolome that identified a panel of gut microbiome-associated serum metabolites, which were used to distinguish adenomas and CRC from healthy individuals [147]. They compared this panel to the use of carcinoembryonic antigen and fecal occult blood tests and found that their panel had improved detection, with up to 83.5% sensitivity and 84.9% specificity [147]. While not yet validated for use in clinical medicine, such results exemplify the promising future of MPE research. Further MPE investigations should be directed with the goal of devising personalize treatment plans and are applicable to many fields of CRC study, including but not limited to identifying personalized biomarkers, understanding the mechanisms of immunotherapy based on the specific tumor and TME, and understanding the effects of diet and medication on the microbiome and CRC risk.

## 4. Preventative Strategies

### 4.1. Diet

As discussed above, certain dietary factors act to promote carcinogenesis through various interactions with the gut microbiome, and it has been proposed that a diet-based strategy for the prevention of CRC could be feasible [148,149].

O’Keefe et al. performed a food exchange study between a high-risk group (African Americans who typically ate a high-fat and low-fiber Western diet), and a low-risk group (rural Africans, who typically ate a low-fat and high-fiber African diet) [35]. The two groups swapped diets for 2 weeks under close supervision. Their study demonstrated that in both the high-risk and low-risk groups, eating a Western-style diet increased mucosal inflammation and proliferation [35]. This intervention was also associated with an increase in fecal *Fusobacterium nucleatum* abundance. Mehta et al., therefore, proposed that *Fusobacterium nucleatum* may serve as a link between diet and CRC development and performed a prospective observational study in which they classified patients into those consuming a prudent diet (whole grains, fruits, and vegetables) and those consuming a Western diet (red and processed meats, desserts, and refined grains) [150]. They followed these patients for 20–30 years and in each diagnosed case of colorectal cancer, tumor samples were studied for the presence of *Fusobacterium nucleatum.* Interestingly, there was an association between *Fusobacterium nucleatum*-positive CRC and diet only, suggesting that perhaps high-fiber diets may impact the risk of CRC through this specific species, although causality was not proven [150]. A similar and more recent study from the same group demonstrated that the low-fiber Western diet is also linked with a higher incidence of high-abundance colibactin-producing *Escherichia coli* CRC [151].

Dietary fibers, additionally, are thought to contribute to colonic mucosal health by providing colonic bacteria with substrates for fermentation [152]. Fermentation of fiber releases SCFAs, which as discussed previously, could possibly serve as mediators against inflammation when kept within a certain range. A recent review, however, found that the majority of studies, from 1989 to 2019, concluded that the administration of such dietary fibers alone is ineffective for preventing CRC [153]. However, it has been suggested that a high fiber intake from consuming whole grains may also indirectly protect against CRC by reducing the incidence of risk factors for CRC, such as obesity and diabetes [154]. Again, these inconsistencies demonstrate that further investigations are needed to elucidate the role of dietary fibers in CRC prevention. 

Other components of dietary habits may also be potential targets for preventative dietary modification. Diets high in fat and red meats are a proven risk factor for CRC development [29,155,156]. High-fat diets have been shown to increase the production of carcinogenic secondary bile acids [29], and bacterial processing of the amino acids obtained through the consumption of red meats can lead to the production of potential carcinogens such as heterocyclic aromatic amines, heme, and N-nitroso compounds [157]. There is also the concept of high-fat diet dysbiosis, in which bacteria of the Firmicutes phylum increase, and bacteria of the Bacteroidetes phylum decrease [158], suggesting a departure from the composition of healthy commensals. 

An interesting study by Shen et al. examined the difference in dietary habits and microbiome composition across 200 patients, half with CRC and half without [159]. Consistent with the literature, healthy patients had a higher intake of fish, dairy products, vegetables, beans, and nuts, while patients with CRC had a higher intake of red meats. About a quarter of the patients in each group submitted fecal samples for microbiome analysis. Higher levels of Proteobacteria and Fusobacteria, which were related to lower intakes of fruits, nuts, and beans, and higher intakes of picked food, were found to be associated with CRC. Lower levels of Firmicutes were also associated with CRC [159], which interestingly, is in opposition to what is seen in high-fat diet dysbiosis.

As shown by Shen et al. [159], fish has been associated with decreased risks of CRC [160], possibly due to the high levels of unsaturated fatty acids, which are known to be anti-inflammatory [156]. Diets rich in antioxidants [161] and dairy have also been shown to be inversely related to the risk of developing CRC, the latter of which is likely due to its high calcium content [162,163,164]. There is mixed evidence regarding the association between CRC and different subsets of dairy intake, such as milk with different fat contents, cheese, yogurt, butter, and creams. This may be due to the presence of various other substances within these dairy products that could potentially increase colorectal cancer risk, such as bile acids, saturated fats, flavorings, and processed sugars [165].

Evidence regarding the consumption of poultry and eggs is inconclusive [165]. Fruits and vegetables, although high in antioxidants, also have not been consistently proven to decrease colorectal cancer risk [166,167]. The literature has areas of inconsistency regarding diet, the microbiome, and CRC risk. There seems to be a consensus that the Western diet is “unhealthy” and associated with a higher incidence of CRC [168], but further study, particularly MPE research, is needed to determine the effects of specific dietary changes and how these changes can be implemented to decrease the risk of CRC. These future studies should attempt to fill the current research gaps with an emphasis on collecting comprehensive microbiome and dietary data in patients well-before and long after a diagnosis of CRC [169].

### 4.2. Probiotics

Probiotics, which encompass any group of live microorganisms that provide health benefits when consumed in adequate amounts [170], have recently been examined in the context of CRC prevention and treatment. The most common probiotics in use include *Lactobacillus* and *Bifidobacterium* species [171]. *Lactobacillus fermentum* and *Lactobacillus casei* have been shown to exert anti-proliferative and pro-apoptotic effects on CRC cells [171,172]. A recent study also demonstrated that *Bifidobacterium* has the ability to inhibit the growth of cancer cells, possibly by decreasing intestinal pH, inhibiting the growth of pathogenic and tumorigenic bacteria [173], thus maintaining intestinal microbiome balance [174]. Additionally, both *Bifidobacterium* and *Lactobacillus* species have been shown to strengthen tight junctions in the intestinal wall, thus promoting epithelial integrity [175]. All this data taken together indicates that probiotics may be a useful adjunct to diet modification in attempts at modifying risk factors for CRC. Some have also proposed that if the detection of biofilms can be worked into the screening process, the use of probiotics may be able to eliminate them [45].

### 4.3. Aspirin and Other Medications

The United States Preventative Services Task Force recommends using aspirin in select populations without a known history of cardiovascular disease or increased risk of bleeding to prevent CRC. The recommendation strength varies with age group [176]. When aspirin was first recommended as chemoprophylaxis, the mechanism was unknown. However recent studies have suggested that the chemoprophylactic effects may be mediated by the effects of salicylic acid, the primary metabolite of aspirin, on the bacteria of the host microbiome, possibly through the inhibition of growth and modulation of their expressed virulence factors [177,178].

Brennan et al. evaluated the effects of aspirin on *Fusobacterium nucleatum,* enterotoxigenic *Bacteroides fragilis,* and colibactin-producing *Escherichia coli*, all of which are known to be associated with CRC. Their results from studying mouse models and human samples suggest that aspirin has direct effects against *Fusobacterium nucleatum* strains, which may be of potential use for higher-risk patients with high-abundance *Fusobacterium nucleatum* colonic adenomas [178]. Studies have also demonstrated that only a 2-year period of aspirin use can significantly reduce the risk of CRC in patients with Lynch Syndrome [179]. The effects in *Escherichia coli* and *Bacteroides fragilis* were also statistically significant but to a much lesser degree than the sensitivity seen in *Fusobacterium nucleatum*, suggesting limited therapeutic benefit [178].

Because bacteria can develop antibiotic resistance, there is interest in evaluating non-antibiotic drugs that have modulating effects on the microbiome in regard to CRC prevention. There are many such drugs that have been shown to influence the development of CRC, including proton pump inhibitors [180], other non-steroidal anti-inflammatory drugs [181], and certain antihistamines [182]. These drugs have also been shown to modulate the microbiome [177,183]. This represents another understudied but valuable area that would benefit from further research.

### 4.4. Fecal Microbiota Transplantation

The most radical therapy for the alteration of the gut microbiome, fecal microbiota transplantation (FMT), is currently a recognized treatment for recurrent *Clostridium difficile* infection, and may also have a role in CRC prevention. Studies in recurrent *Clostridium difficile* infections have shown that FMT can decrease the elevated levels of procarcinogenic colibactin-producing *Escherichia coli* in the colon when transplant samples are negative for this specific strain [184]. In the mouse model, researchers have shown that FMT from human CRC survivors who consume rice bran, an anti-carcinogenic dietary modification, reduces tumor burden [185]. FMT has also been shown to reduce colitis in mice [186], and therefore, is being studied as a method to restore a healthy microbiome and treat inflammatory bowel disease, which is a known risk factor for CRC [187].

The use of FMT, as a commonly practiced method of CRC prevention, faces several challenges, however. FMT is not nearly as easy to incorporate as other methods of microbiome alteration, such as the consumption of probiotics. Furthermore, there are several questions that still need to be addressed, such as what the optimal microbiome for CRC prevention is, whether there is a specific patient population that preventative FMT should be targeted towards, and how to identify an appropriate donor microbiome. As has been discussed throughout this paper, the nature of specific microbes within fecal material and how they interact to influence host health are still under investigation. The ideal microbiome composition has yet to be established, in part because the microbiome and its interactions are so complex. For example, metabolic diseases can affect the microbiome, and there are reports of patients becoming obese after successful treatment of recurrent *Clostridium difficile* infection with FMT from an obese donor [188]. Antibiotic resistance in the form of the fecal resistome, the genes that encode antibiotic resistance within a sample, can also be transferred. Screening for potential pathogens is complicated by the fact that there may not be effective screening tests, such as for those colonized but not infected by *Clostridium difficile* [189]. Additionally, certain microbes may play different roles across individuals or populations, which could influence whether any specific donor would be a good match for a recipient. A good example of this is seen in foregut *Helicobacter pylori*, which is associated with varying degrees of risk for reflux, gastric cancer, and peptic ulcer disease depending on the strain and population in question, although it can also be protective against gastroesophageal cancer [190]. Similar variations are likely present in colonic gut commensals and pathogens; these variations need to be examined to better understand what characteristics are present in the ideal donor specimen. Nonetheless, FMT remains a promising area of study for colorectal cancer prevention as research in this field continues to progress rapidly.

## 5. Conclusions

As one can imagine, the mechanisms of microbiome involvement in CRC pathogenesis are complexly interwoven, and we are only just starting to understand their interplay. Even so, much of what we currently do know is applicable to improving the management of CRC, and many studies have shown promising results, with great strides being made especially in the field of immunotherapy. More research is needed in nearly all aspects of this field, but if we continue on this current trajectory, there will be significant improvements in our ability to prevent, diagnose, and treat CRC of all stages.

## Figures and Tables

**Figure 1 cancers-14-05685-f001:**
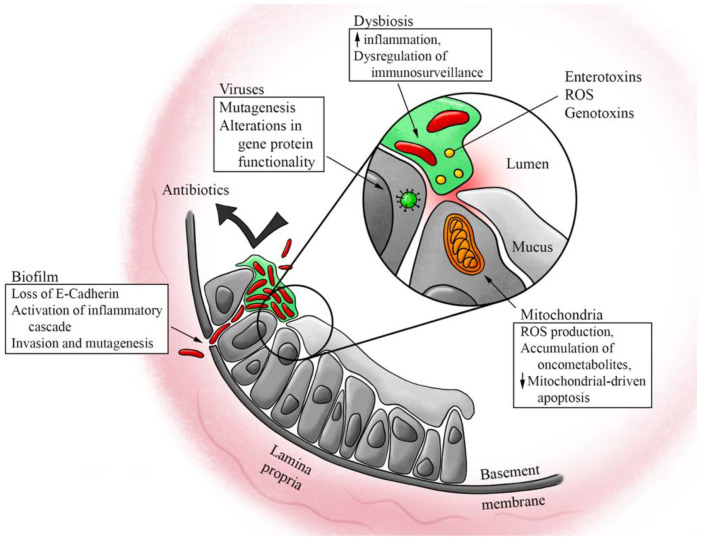
Effects of the microbiome on the pathogenesis and progression of colorectal cancer. Multiple different aspects of the microbiome have been proposed to play a role in the pathogenesis and progression of colorectal cancer, including dysbiosis of the bacterial and viral microbiomes, the biofilms and toxins associated with bacterial dysbiosis, and microbe-mitochondrial crosstalk altering mitochondrial function.

**Figure 2 cancers-14-05685-f002:**
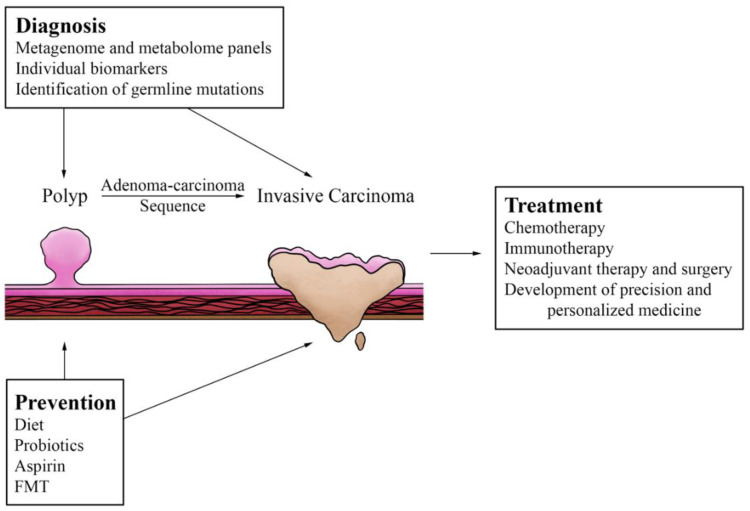
Preventative, diagnostic, and therapeutic options that influence and are under the influence of the microbiome. Interventions such as chemotherapy and aspirin prophylaxis have been shown to influence or be influenced by certain aspects of the microbiome. Research continues to provide or suggest newer management strategies as well, such as use of immunotherapy in colorectal cancer, development of personalized biomarkers, and consideration of fecal microbiota transplant (FMT), all of which also have complex interactions with the gut microbiome.

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
