# Peer review of "The Role of the Microbiome on the Pathogenesis and Treatment of Colorectal Cancer"

_cancers, 2022, doi:10.3390/cancers14225685_

Round 1

Reviewer 1 Report

 The authors wrote a quite interesting review on the microbiota and colorectal cancer (CRC). This is generally of high interest. This covers good amounts of data although it is weak in discussion on some areas. Literature search is inadequate in several areas.

There are big research gaps. The authors fail to describe adequately. First, there are studies that examined foods / nutrients in relation to dysbiosis / pathogenic microbes. The authors touch on studies by O’Keefe’s group. Second, there are studies that examined foods / nutrients in relation to CRC risk. References in this area seem lacking. Third, there are studies that examined differences in stool microbiota between CRC patients and controls. Seminal meta-analysis studies in Nature Medicine 2019 (Wirbel et al.; Yachida et al.) are not mentioned in this area.

In any case, it seems that these study areas do not well connect with each other in many studies. There are indeed gaps. Molecular pathological epidemiology studies that examined foods/nutriest, CRC risks, and pathogenic microbes in tumors nicely can connect these gaps. See a recent study by Arima et al. Gastroenterology (2022) and the classic one by Mehta et al. JAMA Oncol (2017). There are no other similar studies yet, but it will open new research approaches. However, this type of approach is rarely taken. There is a widely open opportunity that is currently missed. These facts should be discussed.

There are also several experimental studies by Garrett’s group, which are not mentioned but important. For instance, a link between aspirin intake and CRC pathogenesis through Fusobacterium nucleatum (Brennan et al. mBio 2021). Common medications (not necessarily cancer treating drugs) make another big topic.

There are also influences of germline genetic variations on microbiome and cancer. This is understudied. But gene-by-environment interactions for CRC pathogenesis can / should be discussed. This topics is neglected. But it is important in early-onset CRC, where the microbiome play a considerable role. See Archambault et al. J Natl Cancer Inst 2022; Akimoto et al. Nat Rev Clin Oncol 2021; Hofseth et al. Nat Rev Gastro Hepatol 2020.

In these above contexts, the authors should discuss gaps and opportunities such as research on dietary / lifestyle factors, microbiome, and personalized molecular biomarkers, which is needed for further research. The authors should discuss molecular pathological epidemiology research that can investigate diet, microbiota, and other factors in relation to molecular pathologies and clinical outcomes. Molecular pathological epidemiology research can be a promising direction (eg, Gut 2022; Annu Rev Pathol 2019; Curr Colorectal Cancer Rep 2017; etc.) and should be discussed in this paper.

There are a number of areas that lack adequate referencing, despite their citing their own references (refs 56-59). Refs 58-59 (on HCC and breast cancer by their own studies) are irrelevant since TAMs have shown different phenotypes by different tumor types. For instance, for M1 and M2 macrophages, there are much better (or larger) human tissue studies (eg, there is one in Cancer Immunol Res 2021) than quoted refs. Please conduct more comprehensive literature search throughout.

Also a quite important paper on this topic is missing (Murray PJ, et al. Macrophage activation and polarization: nomenclature and experimental guidelines. Immunity. 2014;41:14–20).

This is just a tip of iceberg. There are many other places.

Reviewer 2 Report

Yu et al. provides an overview of the associations between the gut microbiome and colorectal cancers. This is a well-written review that is of a depth and complexity appropriate for a broad audience. It was enjoyable to read. I have only minor comments.

Line 86, revise “unfavorable altered” to “unfavorably-altered”

In Lines 86-87, the authors introduce the concept of dysbiosis as a maladapted gut microbiome composition. It is still very difficult to define what a “dysbiotic” gut microbiome is, compositionally, without prior knowledge of host health. It can be easily misleading for less knowledgeable readers to presume that the presence or absence of bacterial species always means disease. In truth these correlations are only correlations and not absolute certainties. I suggest that the authors provide a bit more depth into their description of dysbiosis and the caveats associated with correlations between microbiome composition and disease

Line 280, revise “expensive” to “expense”

In Line 323, the authors describe “immunogenic bacteria”. It would be beneficial to the readers if there were a greater description on the types of bacteria considered immunogenic

In Lines 400-407, the authors summarize studies related to a healthy human blood microbiome. Search of the literature suggests that this is still a debated area of study. One Biorxiv preprint suggests there is no association (

In the FMT section beginning on line 473, an added challenge to using FMTs regularly is the undefined nature of donor material. I suggest that the authors describe the challenges of defining an appropriate donor microbiome.

Reviewer 3 Report

Overall, this is a very well written review article on the role of the gut microbiome with respect to pathogenesis and treatment of colorectal cancer. The text is well structured and special emphasis is given to the interaction between the microbiome and the tumor microenvironment. In the paragraph on “Neoadjuvant Immunotherapy and Surgery” results from the of the NICHE and NICHE2 trials could be added (Coukos et al. Nature Medicine. 2020;26:473-4. doi:10.1038/s41591-020-0826-3 and Chalabi et al. Annals of Oncology (2022) 33 (suppl_7): S808-S869.

The number of own publications cited within this review article is extensive. In some of the cases, citation of more general review articles is suggested.

Example: “Immune checkpoint inhibitors (ICIs) are a form of immunotherapy that blocks the 356 tumor cells from downregulating the immune response against them [113].” 113. Tokumaru, Y.; Joyce, D.; Takabe, K., Current status and limitations of immunotherapy for breast cancer. Surgery 2020, 167 (3), 800 628-630.

A more general review article related to the topic would for example be: Pardoll. The blockade of immune checkpoints in cancer immunotherapy. Nature reviews Cancer. 2012;12:252-64. doi:10.1038/nrc3239.

Round 2

Reviewer 3 Report

With the changes made, the review article is now ready for publication.